# Contemporary Management of Severely Calcified Coronary Lesions

**DOI:** 10.3390/jpm12101638

**Published:** 2022-10-03

**Authors:** Natthapon Angsubhakorn, Nicolas Kang, Colleen Fearon, Chol Techorueangwiwat, Pooja Swamy, Emmanouil S. Brilakis, Aditya S. Bharadwaj

**Affiliations:** 1Division of Cardiology, Loma Linda University Medical Center, Loma Linda, CA 92354, USA; 2Department of Medicine, Loma Linda University Medical Center, Loma Linda, CA 92354, USA; 3Division of Cardiology, University of California, Riverside, CA 92521, USA; 4Minneapolis Heart Institute, Abbott Northwestern Hospital and Minneapolis Heart Institute Foundation, Minneapolis, MN 55407, USA

**Keywords:** coronary artery calcification, intravascular imaging, percutaneous coronary intervention

## Abstract

Coronary artery calcification is increasingly prevalent in our patient population. It significantly limits the procedural success of percutaneous coronary intervention and is associated with a higher risk of adverse cardiovascular events both in the short-term and long-term. There are several modalities for modifying calcified plaque, such as balloon angioplasty (including specialty balloons), coronary atheroablative therapy (rotational, orbital, and laser atherectomy), and intravascular lithotripsy. We discuss each modality’s relative advantages and disadvantages and the data supporting their use. This review also highlights the importance of intravascular imaging to characterize coronary calcification and presents an algorithm to tailor the calcium modification therapy based on specific coronary lesion characteristics.

## 1. Introduction

The prevalence of moderate to severe calcification in coronary lesions being treated with percutaneous coronary intervention (PCI) is between 18 to 24%, according to recent meta-analyses and multiethnic registries [1,2,3]. Advanced age, diabetes mellitus, hypertension, hyperlipidemia, smoking, and chronic kidney disease are associated with coronary calcification [4]. Due to increasing age and comorbidities of patients undergoing PCI, the prevalence of severely calcified coronary lesions is increasing [5]. Severe coronary calcification is independently associated with increased major adverse cardiac events following PCI [2,5]. In addition to long-term adverse outcomes, treatment of calcified coronary lesions also poses significant technical challenges. It is associated with an increased likelihood of procedural failure (such as balloon uncrossability or stent under-expansion), complications (such as coronary dissection, coronary perforation, or balloon rupture), and periprocedural mortality and morbidity [5,6]. The periprocedural assessment of the extent and thickness of coronary calcium is critical for calcium modification planning [7,8]. There are many technologies available to modify severely calcified plaques, such as non-compliant (NC) balloons, rotational, orbital and laser atherectomy, and intravascular lithotripsy (IVL) [9]. Each of these modalities of calcium modification has advantages and disadvantages. The contemporary algorithm for treating severely calcified lesions with a preference for one device over the other is changing, especially with the advent of IVL. In this article, we highlight the intravascular imaging-based characterization of coronary calcification and all the modalities available for calcium modification. We will also review selected relevant clinical trials that support their clinical use, as depicted in Table 1. However, it is to be noted that many of these clinical trials lack hard clinical end points and have focused on stent-related outcomes. Additionally, the comparison of outcomes across clinical trials should be performed with caution given differences in study population and the inherent heterogeneity in the construct of these calcium modification tools. Finally, further investigation of the results in larger contemporary cohorts is needed.

## 2. Definition and Characterization of Coronary Calcification 

Several imaging modalities can identify and characterize calcified coronary lesions, including coronary angiography, coronary CT angiography, and intravascular imaging [33]. Coronary CT angiography has emerged as a useful non-invasive tool to identify coronary calcium and plan coronary interventions. Measurement of coronary artery calcium score can be used to stratify cardiovascular risk as it is a powerful predictor of atherosclerotic cardiovascular disease [5]. Coronary angiography generally demonstrates severely calcified lesions as radiopacities without cardiac motion before contrast injection, frequently visible on both sides of the arterial lumen (tram-track). Intravascular ultrasound (IVUS) enables full-thickness visualization of the coronary artery wall, allowing a detailed evaluation of calcified lesions and deposits within deeper layers of the coronary artery wall. Calcium appears as a bright, hyperechoic arch with acoustic shadowing. Optical coherence tomography (OCT) uses infrared light to create even higher resolution images, with a particular advantage in accurate visualization of calcium thickness. Calcium appears as low-intensity signal areas with well-delineated borders.

The 2021 American College of Cardiology/American Heart Association/Society for Cardiovascular Angiography and Interventions (ACC/AHA/SCAI) guidelines recommend using intracoronary imaging for procedural guidance in complex coronary artery stenting cases (class 2a recommendation, level of evidence B) [8]. Both OCT and IVUS can identify, localize, and quantify coronary artery calcium, allowing a comprehensive pre-PCI assessment of coronary calcium patterns and severity to predict successful stent expansion. Three essential OCT-derived parameters of coronary calcification predicted stent underexpansion, including an arc of calcium ≥ 180°, calcium length > 5 mm, and calcium thickness ≥ 0.5 mm [34]. On IVUS, the length of superficial calcium > 270° (≥5 mm), circumferential 360° calcium, a calcified nodule, and a small caliber vessel (<3.5 mm) predicted stent underexpansion [35]. Calcium scoring systems were developed to identify lesions that may require calcium modification. An OCT-based calcium score of ≥4 or an IVUS-based calcium score of ≥2 was associated with a significantly higher risk of stent underexpansion and indicates the need for calcium modification [34,35]. Table 2 shows a simplified system to categorize calcified coronary lesion severity into mild/moderate/severe, based on the presence of high-risk features on intravascular imaging [36]. 

## 3. Modalities for Coronary Calcium Modification

### 3.1. Balloon Angioplasty

#### 3.1.1. Non-Compliant Balloons

Balloon angioplasty (BA) with NC balloons is generally effective in modifying mildly to moderately calcified coronary lesions to allow more optimal stent expansion. NC balloons can tolerate high pressures and allow more uniform balloon expansion than semi-compliant balloons. However, in severely calcified lesions, there may be non-uniform balloon expansion causing complications such as coronary dissection, coronary perforation, or balloon rupture due to dog-bone deformation exerting excessive pressure at the edges [37]. BA with NC balloon is however a useful adjunct to atherectomy and should always be performed following debulking with atherectomy, in order to ensure that adequate plaque modification has been achieved before stent implantation. 

#### 3.1.2. High-Pressure Balloons

The super high-pressure balloon (OPN NC, SIS Medical, Frauenfeld, Switzerland) is a rapid-exchange NC balloon catheter with twin-layer technology that tolerates very high pressures (up to 35 atm) with minimal increases in diameter within the balloon [38,39,40,41]. In a recently published retrospective study of 326 patients, the super high-pressure balloon successfully treated >90% of non-dilatable calcified coronary lesions in which conventional NC balloons failed to achieve adequate minimal luminal area. Coronary rupture occurred in 3 patients [42]. The super high-pressure balloon can also be used for post-dilation for optimal stent expansion [43]. Although this device seems promising, one must consider the potential risk of coronary dissection and perforation.

#### 3.1.3. Cutting Balloons

The cutting balloon (FlexTome and Wolverine, Boston Scientific, Marlborough, MA, USA) is an NC balloon catheter with three or four atherotomes attached longitudinally along the outer balloon surface. When the balloon is inflated, the blades create shallow incisions into the calcified atherosclerotic plaque to improve stent expansion [13]. The microsurgical blades also assist with anchoring the balloon in place, which is especially helpful for ostial or in-stent restenosis (ISR) lesions [11]. The rigidity of the blades can cause difficulty in delivering the device to the target lesion, which is a significant limitation; however, a recent study suggested a better crossability profile of the new cutting balloon model (Wolverine, Boston Scientific, Marlborough, MA, USA) when compared with scoring balloons [44]. Reported complications include coronary artery perforation and blade entrapment.

Initial experience with cutting BA in calcified lesions appeared favorable because the acute lumen gain achieved by the cutting balloon was significantly larger than standard BA [10,12,13,45,46]. However, a large, randomized trial showed similar acute procedural success, defined as residual diameter stenosis < 50% without in-hospital major adverse cardiovascular event (MACE), between the cutting balloon group and the conventional balloon group. The perforation rate was, however, higher in the cutting balloon group (0.8% vs. 0%, *p* = 0.03) [10]. More recently, a large retrospective study comparing the use of cutting balloon, BA, and RA in PCI with drug-eluting stent (DES) showed similar mortality, target lesion revascularization (TLR), and MACE across these modalities [47]. Cutting BA is a valuable adjunct in modifying calcified lesions and could mitigate the use of more advanced technology if employed in an appropriate setting.

#### 3.1.4. Scoring Balloons

The scoring balloon (AngioSculpt, Philips, San Diego, CA, USA; Scoreflex, OrbusNeich, Hong Kong, China; Chocolate XD, Teleflex, Wayne, PA, USA; NSE Alpha, B. Braun, Melsungen, Germany; Lacrosse NSE, Asomedica, Minsk, Belarus) is a semi-compliant balloon catheter encircled by sharp scoring elements on the surface, which permits the focal application of the force to the calcified plaque throughout inflation. Although scoring balloons and cutting balloons are mechanistically similar, scoring balloons have a more deliverable profile and are associated with less vessel wall injury and a reduced risk of coronary dissection, with a preserved degree of luminal expansion when compared with cutting balloons [14,15,48]. Although no specific randomized controlled trial compares these two modalities, the scoring balloon has generally been considered an alternative to a cutting balloon.

In a feasibility trial of 60 patients, the AngioSculpt (Philips, San Diego, CA, USA) balloon showed a high procedural success rate with no serious procedural complications, confirming its technical feasibility and safety when applied to de novo or ISR coronary lesions prior to delivering DES [49]. Although scoring balloons were not initially intended for severe coronary calcification, preliminary experience has revealed their practicability for use during PCI [50,51,52]. In one study, scoring BA led to successful calcium modification of severely calcified lesions in 68%, although there was no control group in this study [53]. 

In the PREPARE-CALC trial, 200 patients with severely calcified coronary lesions undergoing PCI were randomized to either RA or modified BA (cutting and scoring balloons). While strategy success was greater in the RA group, at nine months, there was no significant difference between the two groups in late lumen loss, TLR, or target vessel failure [24]. In the authors’ opinion, modified BA with either cutting or scoring balloons remains a reasonably effective and safe form of plaque modification and has a role in moderately calcified lesions, fibrotic lesions (such as side-branch ostium), and ISR lesions. Although modified BA with specialty balloons has advanced in recent years, resulting in better deliverability, there is still opportunity for further improvement in design.

### 3.2. Intravascular Lithotripsy

Adapted from the lithotripsy technology for treating nephrolithiasis, IVL (Shockwave C2 coronary IVL, Shockwave Medical, Santa Clara, CA, USA) is the newest addition to the armamentarium for modification of severe coronary calcification. The coronary IVL balloon catheter is 12 mm long and available in diameters from 2.5 to 4.0 mm. The balloon is sized with a ratio of 1:1 with the reference coronary diameter. When in place, the balloon is inflated to 4 atm to allow vessel wall apposition, and then impulses of mechanical energy are delivered at a frequency of 1 pulse per second, for ten pulses in sequence for a maximum of 80 pulses per balloon. These shockwaves generate a peak positive pressure of up to 50 atm with less than 5 microseconds in duration, causing a vibration that selectively cracks calcified areas within the superficial and deep layers of the vessel while sparing soft tissue due to its elasticity [54].

The effects of this novel technique on calcified coronary lesions were first reported in 2017. Following IVL, improvement in vessel compliance (related to circumferential calcium fractures), luminal gain, and stent expansion have been demonstrated by OCT, while no significant complication was reported [55]. Circumferential calcium fracture was observed, and IVL-induced fractures were independent of the thickness of calcium; in fact, fractures occurred more frequently as the severity of the calcification increased [55]. New data from recent single-arm studies of IVL are promising, with a high procedural success rate and low risk of significant complications. The Disrupt CAD I study was the first single-arm multicenter study that showed IVL was feasible and facilitated the delivery of stents to all target lesions with moderate or severe calcification. On average, stenosis was reduced to 12% with an acute luminal gain of 1.7 mm. The use of IVL was safe and without any dissection, slow flow or no reflow event, embolization, or coronary perforation [29]. Following this pilot study, Disrupt CAD II [30], Disrupt CAD III [31], and Disrupt CAD IV [32] studies showed promising outcomes for IVL in severely calcified coronary lesions. A patient-level pooled analysis of Disrupt CAD studies (628 patients enrolled at 72 sites from 12 countries) showed a primary safety endpoint (freedom from 30-day MACE) of 92.7% and an effectiveness endpoint (procedural success, defined as stent delivery with residual stenosis ≤ 30% by quantitative coronary angiography without in-hospital MACE) of 92.4% [56]. 

IVL offers several advantages. First, while calcium modification by OA and RA can generate tiny microparticles that may embolize distally and impair coronary circulation, in IVL, calcium fragments remain within subintima with minimal intimal disruption; therefore, IVL is less likely to cause distal embolization [55]. Compared with an incidence of slow flow/no reflow event of up to 2.5% with RA, slow flow/no reflow event was not observed with IVL in any Disrupt CAD studies [29,30,31,32,57]. Second, because the device is delivered similar to standard catheter-based PCI, IVL requires no specific training compared to traditional atherectomy and has no learning curve for the operator. Third, in contrast to atherectomy, IVL is not subject to guidewire bias. The mechanical energy from IVL is distributed uniformly across the inflated balloon into both superficial and deeper layers of the vessel, addressing calcium irrespective of circumferential location and depth [58,59]. Lastly, clinical experience has shown that IVL may also be used to modify calcium in ISR or underexpanded stents implanted in severely calcified lesions [60,61,62,63], including freshly deployed stents [64]. Moreover, IVL may be used as an adjunct to other atherectomy techniques when there has been inadequate calcium modification [65].

The acoustic shockwaves of IVL can induce ventricular or atrial ectopic beats and asynchronous cardiac pacing. These transient IVL-induced captures were commonly observed during impulse generation; however, no ventricular tachyarrhythmias or adverse clinical outcomes have been reported [66]. Although the IVL balloon catheter is less deliverable than a standard NC balloon catheter, 6-French guide extensions can usually facilitate device delivery. Although an OCT substudy from Disrupt CAD shows that IVL is safe and effective in the treatment of eccentric calcium, its role in the management of calcified nodules needs further evaluation [67].

### 3.3. Coronary Atherectomy

#### 3.3.1. Rotational Atherectomy

RA (Rotablator and RotaPro, Boston Scientific, Marlborough, MA, USA) utilizes a high-speed rotating diamond-tipped burr to mechanically ablate hard calcified atheroma while deflecting off pliable noncalcified tissue. The high-speed burr rotation causes lumen enlargement with a smoother luminal surface and reduced plaque rigidity, enabling balloon predilatation and increased stent expansion [68]. Current guidelines on the management of calcified coronary lesions state that RA can be beneficial to improve procedural success for fibrotic or heavily calcified lesions (class 2a, level of evidence B) [8]. 

RA has been investigated in various clinical settings and is considered the gold standard for modifying severely calcified lesions prior to stenting, especially in balloon uncrossable lesions [16,17,18,19,20,21,22]. Historically, RA was associated with higher procedural complication rates in comparison to standard PCI, with an incidence of 9.7% in one large-scale study [69]. These complications include coronary dissection, perforation, burr entrapment, wire fracture, and atrioventricular block requiring pacemaker insertion. During RA, atherosclerotic debris from the calcified lesions is released into the coronary circulation, which can potentially cause transient slow or no reflow [70]. The occurrence of these complications may reflect the lesion complexity rather than the device issues themselves, and it may be minimized by the contemporary technique, which has remarkably improved in recent years [57,69]. Modern approaches to mitigate the risk of complications include the use of a smaller burr with a lower burr to artery ratios, gradually advancing the burr in small increments, shorter burring episodes, avoiding burr deceleration > 5000 rpm, allowing for adequate time intervals between burring, and avoiding extreme tortuosity [57,71]. Centers performing a higher volume of RA procedures had lower incidence of major complications and lower mortality compared with low-volume centers [72,73]. These data reaffirm the safety of RA in the context of substantial improvements in appropriate patient selection, operator experience and their techniques, and center caseload. 

With the advent of DES, more recent studies have been done to evaluate the utility of RA. The ROTAXUS study randomized 240 patients with calcified lesions to RA before stenting or direct stenting [23]. This trial showed a higher rate of procedural success in the RA group (92.5% vs. 83.3%, *p* = 0.03). However, despite greater minimal luminal area with RA, at nine months there was a greater ISR rate, which may have been related to the use of older-generation DES. Rates of MACE, TLR, and definite stent thrombosis were not significantly different between the two groups. However, the trial is limited by a significant crossover rate and the exclusion of more severely calcified lesions. The result from a more recent PREPARE-CALC study suggested that RA had greater procedural success (98% vs. 81%, *p* = 0.0001) when compared with cutting or scoring balloons [24]. Procedural complication rates were low and similar in the two groups. There was no increase in late lumen loss in the RA arm of the PREPARE-CALC study, which used sirolimus-eluting stents. While the main benefit of RA is to facilitate successful PCI of severely calcified lesions, there are currently no convincing data regarding the long-term clinical benefits of RA.

#### 3.3.2. Orbital Atherectomy

OA (Diamondback 360, Cardiovascular systems Inc., St. Paul, MN, USA), was approved by the US Food and Drug Administration in 2013 to treat severely calcified coronary lesions. This device uses the elliptical movement of a 1.25 mm eccentrically mounted diamond-coated crown to create centrifugal force that selectively ablates non-flexible calcified plaque [74].

There are several theoretical advantages of OA, which highlight differences in mechanisms and structural components between OA and RA [75,76]. Whereas the RA burr is ideally advanced with the slow pecking motion, the OA crown is best advanced with a slow, continuous motion with the possibility to slow down in segments that require more ablation. The safety of this technique relates to the elliptical movement of the smaller OA crown and comparatively rapid flow of ViperSlide, facilitating the flush of microparticles through the microvasculature. Moreover, there is less interruption in blood flow during crown orbiting and less vascular heating. These features potentially reduce the likelihood of slow/no reflow events and thermal injury during the procedure. In addition, there is a diamond coating on the entire OA crown permitting bidirectional atherectomy, making OA burr entrapment theoretically less likely than RA [77]. Furthermore, the ability of the OA device to treat the lesion in a retrograde fashion is advantageous and likely safer in aorto-ostial and tortuous lesions.

The ORBIT study series examined use of orbital atherectomy in coronary artery disease. ORBIT I was a prospective, non-randomized study of 50 patients with de novo calcified coronary lesions treated with OA and PCI, reporting a procedural success (defined as residual stenosis < 20% after stenting) in 94% of cases [25]. No cases of slow/no reflow event following OA were documented, and 3-year and 5-year MACE rates were 18.2% and 21.2%, respectively [78]. The pivotal ORBIT II study (prospective, non-randomized, multicenter) reaffirmed the results of the ORBIT I study in 443 patients demonstrating procedural success in 91.4% of cases [26]. The complication rates were low compared with historical controls. Severe coronary dissection and coronary perforation occurred in 3.4% and 1.8% of cases, respectively. The cumulative 3-year MACE rate was 23.5%, and the TLR rate was 7.8% [79].

While current data suggest OA is an effective and safe strategy with acceptable rates of MACE and procedural complications, no randomized controlled trial to date has directly compared the efficacy and safety of OA and RA. Compared with RA, OA performs more profound plaque modification of both superficial and deep calcium as seen on OCT and was associated with a significantly lower rate of stent strut malapposition compared with RA [58]. Another recent observational study suggested that in-hospital MI and in-hospital mortality rates were less frequent with OA compared to RA, and there were no significant differences in procedural complications between the two groups, with OA having a comparable coronary perforation and dissection rate of 0.4% and 1.3%, respectively [80]. The ECLIPSE study (NCT03108456) is an ongoing randomized controlled trial, comparing OA with conventional BA before DES implantation in severely calcified lesions [81].

#### 3.3.3. Laser Atherectomy

ELCA (CVX-300, Philips, San Diego, CA, USA) is a calcium modification technique introduced more than 20 years ago as an alternative to BA. This technique modifies plaque through a process known as photoablation. The device generates pulses of short-wavelength, high-energy ultraviolet light, causing vaporization of water, dissociation of carbon bonds, and molecular vibration, leading to plaque obliteration and greater luminal expansion [82]. Catheters are available in 4 diameters, including 0.9 mm, 1.4 mm, 1.7 mm, and 2.0 mm. The recommended catheter size is based on a catheter/vessel diameter ratio of 0.5–0.6 [83].

Laser atherectomy has several niche indications, including balloon uncrossable or undilatable lesions, chronic total occlusions, debulking vein graft disease, and treating calcific non-dilatable ISR [84,85,86,87,88,89,90,91]. The Laser Veterans Affairs (LAVA) Multicenter Registry evaluated the use of ELCA in 130 target complex coronary lesions performed between 2008 and 2016 [92]. 62% of lesions were de novo moderately or severely calcified lesions and 37% ISR. Use of ELCA was associated with a high technical success rate (90.0%) and procedural success rate (88.8%), and a low MACE rate (3.45%). 

## 4. Management Algorithm for Calcified Coronary Lesions

In this article, we propose an updated algorithm for the management of calcified coronary lesions reflecting contemporary advancements in technology (Figure 1). This novel algorithm begins with assessment of coronary calcium on fluoroscopy (step 1) and intravascular imaging (step 2). The algorithm also describes specific lesion characteristics where one modality of calcium modification may be preferred over another (step 3). Finally, the algorithm also highlights the importance of assessment of successful lesion modification and the adjunctive role of IVL after conventional atherectomy (step 4). The algorithm aims to provide comprehensive yet step-by-step guidance on how to approach calcified coronary lesions. 

Detection and characterization of coronary calcium is an important first step. However, it should be recognized that every modality of calcium assessment has its relative advantages and disadvantages. While coronary angiography has a high positive predictive value for detecting calcification, it has only low to moderate sensitivity compared to IVUS or OCT [93]. Therefore, operators should have a low threshold for performing intravascular imaging to detect calcium, especially in an at-risk population such as those having hypertension, diabetes, renal failure or a history of smoking. Calcium severity (Table 2) and characteristics (intimal vs. medial calcification) should be assessed. The decision regarding which modality of calcium modification to use should be guided based on findings of intracoronary imaging, lesion characteristics, and device availability. The decision should also be guided based on operator experience as higher operator volume is associated with improved PCI outcomes for certain modalities (such as RA) that may require a longer learning curve [94]. OA is preferred in the presence of superficial and deep calcium, long calcified lesions, large caliber arteries, and aorto-ostial lesions. OA is also preferred when there is more than one target lesion to be treated with a discrepancy in reference vessel diameters (e.g., 2.5 mm left anterior descending artery and 4.0 mm left main coronary artery) because of the ability to treat at 80,000 and 120,000 rpm with the same crown and be able to achieve contact with vessel wall for plaque modification. RA or ELCA is preferred in balloon uncrossable lesions. IVL is preferred in the presence of focal severely calcified lesions, bifurcation lesions with severe calcification, and large caliber arteries. IVL or ELCA is preferred in cases of ISR due to an underexpanded stent in the presence of calcium outside the stent. Additionally, if there is still a lack of complete expansion of a 1:1 NC balloon after atherectomy, IVL is recommended to further modify the plaque before stent placement. IVL is expected to become a predominant treatment modality for many calcified coronary lesions in the future due to its ease of use and low risk of major complications associated with other technologies. 

## 5. Conclusions

Coronary artery calcification poses significant challenges during PCI and is associated with increased adverse events both in the short-term and long-term. Intracoronary imaging and determination of coronary calcification severity and characteristics are the keys to guiding further treatment decisions. There are several devices available for the modification of severely calcified plaques, including balloon-based devices, coronary atheroablative options (rotational, orbital, and laser atherectomy devices), and the newer IVL device. Each of these devices is preferred in different scenarios based on coronary lesion characteristics and location as discussed in our algorithm. 

## Figures and Tables

**Figure 1 jpm-12-01638-f001:**
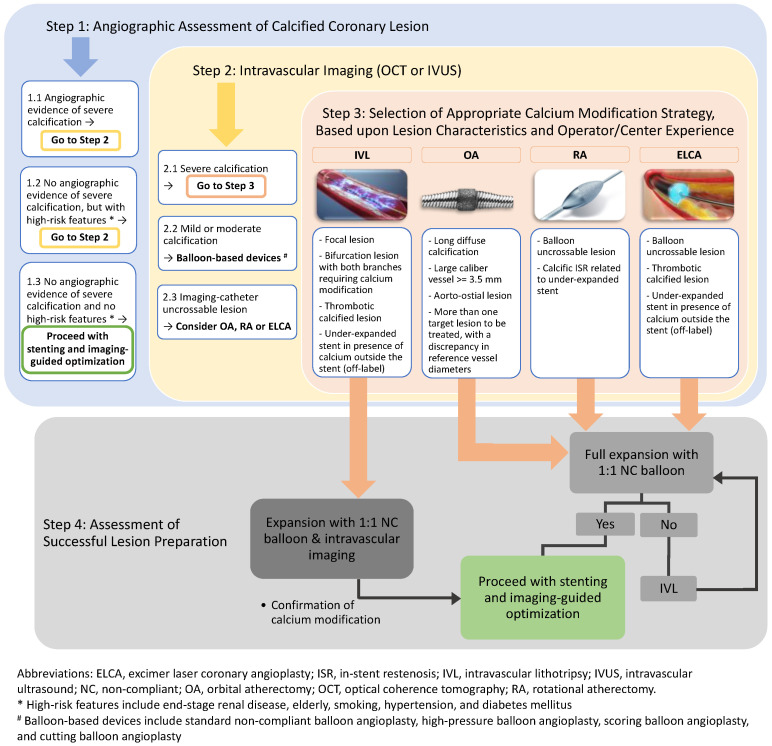
Algorithmic approach for management of calcified coronary lesions.

**Table 1 jpm-12-01638-t001:** Relevant clinical trials for the treatment of coronary calcification.

Study	Study Arms	Relevant Endpoint(s)	Outcomes/Results *	Conclusions
**Cutting Balloon Angioplasty**				
GRT [10]	CBA vs. PTCA	Binary restenosis after 6 months	CBA: 31.4% PTCA: 30.4% *p* = NS	No reduction in restenosis with CBA after 6 months.
REDUCE (unpublished)	CBA vs. PTCA	Binary restenosis after 6 months	CBA: 32.7% PTCA: 25.5% *p* = NS	No reduction in restenosis with CBA after 6 months.
RESCUT [11]	CBA vs. PTCA for ISR	Binary restenosis after 7 months	CBA: 29.8% PTCA: 31.4% *p* = NS	No reduction in recurrent ISR with CBA after 7 months.
CBA before DES [12]	CBA before DES vs. BA	Minimum stent CSA (mm^2^), Acute lumen gain (mm^2^)	CBA:6.26 ± 0.4, 3.74 ± 0.38 BA:5.03 ± 0.33, 2.44 ± 0.29 *p* = 0.031, 0.015	CBA achieved larger lumen CSA and larger lumen gain compared to BA.
Mechanisms of Acute Lumen Gain Following Cutting Balloon Angioplasty in Calcified and Noncalcified Lesions [13]	CBA vs. BA in calcified and non-calcified group	ΔEEM CSA (mm^2^), ΔP + M CSA (mm^2^), Δlumen CSA (mm^2^)	Calcified lesions: CBA: 1.4 ± 1.7, −2.3 ± 1.9, 3.7 ± 1.5 BA: 1.2 ± 1.2, −1.8 ± 1.9, 3.0 ± 1.5 *p* = NS, NS, 0.05 Non-calcified lesions: CBA: 1.0 ± 1.8, −2.9 ± 2.1, 3.9 ± 1.9 BA: 1.6 ± 1.8, −2.0 ± 1.9, 3.6 ± 1.6 *p* = NS(0.11), 0.03, NS	In calcified lesions, CBA achieves a larger lumen gain vs. BA. In noncalcified lesions, there is larger plaque reduction with CBA but no difference in lumen gain vs. BA.
**Scoring Balloon Angioplasty**				
Intimal disruption and cobalt-chromium DES [14]	SBA vs. BA	Stent expansion, lumen eccentricity, intimal disruption frequency, extent	SBA: 68%, 0.94, 68%, 122° BA: 62.1%, 0.80, 0.8, 65° *p* = 0.017, 0.18, 0.035, 0.035	SBA achieved increased stent expansion with similar lumen eccentricity when compared with BA. SBA had more frequent and extensive intimal disruption when compared with BA.
Predilatation with SBA vs. NC [15]	SBA vs. NC	Stent expansion (mm), in-stent late loss after 1 year (mm)	SBA: 70.7 ± 11.2, 0.71 ± 0.63 NC: 69.1 ± 11.1, 0.23 ± 0.52 *p* = NS, 0.03	SBA achieved decreased in-stent late loss when compared to NC after 1 year. There was no difference in stent expansion between SBA and NC groups.
**Rotational Atherectomy**				
ERBAC [16]	RA vs. ELCA vs. PTCA	Procedural success ^∑^, TVR after 6 months	RA: 89%, 42.4% ELCA: 77%, 46% PTCA: 80%, 31.9% *p* = 0.0019, 0.013	RA achieved superior procedural success when compared with ELCA and PTCA, but both RA and ELCA had unfavorable late outcomes when compared with PTCA.
COBRA [17]	RA vs. PTCA	Binary restenosis after 6 months	RA: 49% PTCA: 51% *p* = 0.35	RA did not reduce restenosis after 6 months when compared with PTCA.
DART [18]	RA vs. PTCA in small vessels (2–3 mm)	TVF after 12 months	RA: 30.5% PTCA: 31.2% *p* = 0.98	RA did not reduce TVF after 12 months when compared with PTCA.
STRATAS [19]	Aggressive RA (B/A 0.7–0.9) with PTCA (<1 bar) vs. routine RA (B/A < 0.7) with PTCA (4 bar)	Binary restenosis after 6 months	Aggressive: 58% Routine: 52% *p* = NS	Aggressive RA debulking did not reduce restenosis after 6 months when compared with routine RA debulking.
CARAT [20]	Aggressive RA (B/A > 0.7) vs. Routine RA (B/A = 0.7)	MACE after 6 months	Aggressive: 36.3% Routine: 32.7% *p* = NS	Aggressive RA debulking did not reduce MACE after 6 months compared with routine RA debulking.
ROOSTER [21]	RA (B/A = 0.7) vs. PTCA for diffuse ISR with IVUS guidance	TLR after 9 months	RA: 32% PTCA: 45% *p* = 0.04	RA achieved less TLR after 9 months compared with PTCA in diffuse ISR.
ARTIST [22]	RA (B/A = 0.7) vs. PTCA for diffuse ISR with IVUS guidance in a subset	MACE after 6 months	RA: 80% PTCA: 91% *p* = 0.0052	PTCA achieved a lower MACE when compared to RA in diffuse ISR.
ROTAXUS [23]	RA with DES vs. DES	Late lumen loss (mm) after 9 months	RA with DES: 0.31 ± 0.52 DES: 0.44 ± 0.58 *p* = 0.04	RA before DES achieved increased late lumen loss when compared to DES alone.
Prepare-CALC [24]	RA vs. modified CSA	Successful stent delivery and expansion, late lumen loss (mm) after 9 months	RA: 98%, 0.22 ± 0.41 CSA: 81%, 0.16 ± 0.40 *p* = 0.001, 0.21	RA achieved greater success at stent delivery and expansion than CSA and had similar late lumen loss rates after 9 months.
**Orbital Atherectomy**				
ORBIT I [25]	OA single arm	Device success ^∫^ Procedural success ^∬^ TLR, MACE after 6 months	Device success: 98% Procedural success: 94% TLR, MACE (6 months): 2%, 8%	OA successfully facilitated stent delivery with a low cumulative TLR and MACE after 6 months.
ORBIT II [26]	OA single arm	Safety endpoint ^Ω^ (95% CI) Efficacy endpoint ^Ψ^ (95% CI)	Safety endpoint: 89.6% (86.7–92.5%) Efficacy endpoint: 88.9% (85.5–91.6%)	OA significantly exceeded the primary safety and efficacy endpoints of 83% and 82% respectively. OA also improved in-hospital and 30-day outcomes compared to historic controls with severe CAC.
**Laser Atherectomy**				
LAVA [27]	ELCA vs. PTCA in native vessels or SVG	MACE after 6 months	ELCA: 28.9% PTCA: 23.5% *p* = 0.55	ELCA did not reduce MACE after 6 months compared with PTCA in native vessels or SVG.
AMRO [28]	ELCA vs. PTCA in native vessels	MACE after 6 months	ELCA: 33.3% PTCA: 29.9% *p* = 0.55	ELCA did not reduce MACE after 6 months compared with PTCA in native vessels.
**Intravascular Lithotripsy**				
DISRUPT CAD I [29]	Coronary IVL single arm	Safety endpoint ^Ω^ Effectiveness endpoint ^Ψ^	Safety endpoint: 95% Effectiveness endpoint: 98.5%	Coronary IVL safely and effectively aided stent placement with minimal perioperative complications.
DISRUPT CAD II [30]	Coronary IVL single arm	Safety endpoint ^Ω^ Effectiveness endpoint ^Ψ^ Calcium fractures measured by OCT Mean stent expansion	Safety endpoint: 100% Effectiveness endpoint: 94.2% Calcium fractures: 67.4% Mean stent expansion: 101.7%	Coronary IVL safely and effectively aided stent placement with minimal perioperative complications. OCT demonstrated that calcium fractures were an underlying mechanism for IVL. Coronary IVL allowed for excellent stent expansion.
DISRUPT CAD III [31]	Coronary IVL single arm	Safety endpoint ^Ω^ (lower-bound of 95% CI) Effectiveness endpoint ^Ψ^ (lower-bound of 95% CI)	Safety endpoint: 92.2% (89.9%, *p* = 0.0001) Effectiveness endpoint: 92.4% (90.2%, *p* = 0.0001)	Coronary IVL safely and successfully assisted with stent delivery. The lower bounds of the 95% CI for the safety and effectiveness endpoints exceeded the performance goal of 84.4% and 83.4%, respectively.
DISRUPT CAD IV [32]	Coronary IVL single arm	Safety endpoint ^Ω^: CAD IV cohort vs. propensity matched historical IVL control group Effectiveness endpoint ^Ψ^: CAD IV cohort vs. propensity matched historical IVL control group	Safety endpoint: 93.8% vs. 91.2%, *p* = 0.008 Effectiveness endpoint: 93.8% vs. 91.6%, *p* = 0.007	Coronary IVL safely and effectively aided stent placement with minimal perioperative complications. The results from coronary IVL in the Japanese CAD IV cohort were non-inferior to those from a study of patients treated with IVL in the USA and Europe.

Abbreviations: ΔEEM, change in external elastic membrane; ΔP + M, change in plaque plus media; Δlumen, change in lumen or acute lumen gain; B/A, burr/artery ratio; BA, balloon angioplasty; BMS, bare-metal stent; CABG, coronary artery bypass surgery; CAC, coronary artery calcification; CBA, cutting balloon angioplasty; CI, confidence interval; CSA, cross-sectional area; DES, drug-eluting stent; ELCA, excimer laser coronary angioplasty; ISR, in-stent restenosis; IVL, intravascular lithotripsy; IVUS, intravascular ultrasound; MACE, major adverse cardiac events; MI, myocardial infarction; NS, nonsignificant; NC, noncompliant balloon; OA, orbital atherectomy; OCT, optical coherence tomography; PTCA, percutaneous transluminal coronary angioplasty; PTRA, percutaneous transluminal rotational atherectomy; RA, rotational atherectomy; SBA, scoring balloon angioplasty; SVG, saphenous vein graft; TVF, target vessel failure; TVR, target vessel revascularization. * In order of relevant endpoints; ∑ Diameter stenosis < 50%, absence of death, non-Q-wave MI, or CABG; ∫ Residual stenosis < 50% without device malfunction; ∬ <20% residual stenosis; Ω 30-day freedom from MACE; Ψ residual stenosis < 50% without in-hospital MACE.

**Table 2 jpm-12-01638-t002:** Classification of calcified coronary lesion severity based on intravascular imaging.

Severity	OCT-Based Calcium Score Calcium Arc of >180° (2 Points), 90–180° (1 Point) Calcium Length of >5 mm (1 Point) Calcium Thickness of >0.5 mm (1 Point)	IVUS-Based Calcium Score Length of Calcium (>270°) of >5 mm (1 Point) Presence of 360° Circumferential Calcium (1 Point) Vessel Diameter of ≤3.5 mm (1 Point) Presence of a Calcified Nodule (1 Point)
Mild to moderate	0–3	0–1
Severe	≥4	≥2

Abbreviations: IVUS, intravascular ultrasound; OCT, optical coherence tomography.

## Data Availability

Not applicable.

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
