# Peer review of "Contemporary Management of Severely Calcified Coronary Lesions"

_jpm, 2022, doi:10.3390/jpm12101638_

Round 1

Reviewer 1 Report

The authors have described the different modalities available for assessing coronary calcification and proposed an algorithmic approach for managing these calcified coronary lesions.

Some of my concerns are that in the introduction, the authors have suggested that the prevalence of PCI is 18 to 24%. What is not clear is that 18-24% frequency is limited to the USA or the overall population. In any case, they can provide national and international status. Also, it would help if the authors could compare the prevalence rates with other countries where the prevalence rate is much higher.

The authors state that “Due to increasing age and comorbidities of our patient population,” does “our” indicate the entire population or specific to the patient population they treat?

In the Introduction, the authors have stated that “It is associated with an increased likelihood of procedural failure, complications, and periprocedural mortality and morbidity “It will help if the authors explain the type of procedural failures and what they mean by complications.

 Although, the authors have quoted an article that has compared different techniques used in treating calcified lesions. It is still important to introduce why the preference for one device (IVL) over the other is changing.

 Not straightforward how their proposed algorithm for the modality of calcium modification will use operators’ experiences.

Although they have reviewed each technique individually, they can still detail how their proposed algorithm will improve the current practice.

They have explained how the different technique is chosen depending on the type of lesion. But not apparent why the IVL has been used more than other methods

Author Response

Point 1: Some of my concerns are that in the introduction, the authors have suggested that the prevalence of PCI is 18 to 24%. What is not clear is that 18-24% frequency is limited to the USA or the overall population. In any case, they can provide national and international status. Also, it would help if the authors could compare the prevalence rates with other countries where the prevalence rate is much higher. 

Response 1: Thank you. We have reviewed the three studies cited that evaluated prevalence of coronary calcification during PCI. The first one is a large multinational study (n = 6296), and the other two were done in the USA and the Netherlands, respectively. As they found similar rates of coronary calcification found during PCI (around 20%, 18%, and 19%, respectively), we have made changes to the text.

Point 2: The authors state that “Due to increasing age and comorbidities of our patient population,” does “our” indicate the entire population or specific to the patient population they treat?

Response 2: “Our” was used to indicate patients undergoing PCI. This has been clarified in the manuscript. 

Point 3: In the Introduction, the authors have stated that “It is associated with an increased likelihood of procedural failure, complications, and periprocedural mortality and morbidity “It will help if the authors explain the type of procedural failures and what they mean by complications.

Response 3: We agree with the reviewer’s suggestion. The sentence was modified as following: “It is associated with an increased likelihood of procedural failure (such as balloon uncrossability or stent under-expansion), complications (such as coronary dissection, coronary perforation, or balloon rupture), and periprocedural mortality and morbidity.”

Point 4: Although, the authors have quoted an article that has compared different techniques used in treating calcified lesions. It is still important to introduce why the preference for one device (IVL) over the other is changing.

Response 4: Authors completely agree with reviewer’s comments and the following sentence has been added: “IVL is expected to become a predominant treatment modality for many calcified coronary lesions in the future due to its ease of use and low risk of major complications associated with other technologies.” 

Point 5: Not straightforward how their proposed algorithm for the modality of calcium modification will use operators’ experiences.

Response 5: Thank you for this recommendation. One additional reference and the following sentence have been added to the manuscript: “The decision should also be guided based on operator experience as higher operator volume is associated with improved PCI outcomes for certain modalities (such as RA) that may require a longer learning curve [94].”

Point 6: Although they have reviewed each technique individually, they can still detail how their proposed algorithm will improve the current practice.

Response 6: We agree with the reviewer’s suggestion and the following sentences have been added: “This novel algorithm begins with assessment of coronary calcium on fluoroscopy (step 1) and intravascular imaging (step 2). The algorithm also describes specific lesion characteristics where one modality of calcium modification may be preferred over another (step 3). Finally, the algorithm also highlights the importance of assessment of successful lesion modification and the adjunctive role of IVL after conventional atherectomy (step 4). The algorithm aims to provide a comprehensive, yet step-by-step guidance on how to approach calcified coronary lesions.” 

Point 7: They have explained how the different technique is chosen depending on the type of lesion. But not apparent why the IVL has been used more than other methods.

Response 7: While we believe that most modalities can be used in most situations, we have highlighted some specific instances where one technology may be preferable over another. We have also inserted the following line: “IVL is expected to become a predominant treatment modality for many calcified coronary lesions in the future due to its ease of use and low risk of major complications associated with other technologies.”

Reviewer 2 Report

Well written review Remarks - Obviously a review collects the strengths and weaknesses of the available studies. If the studies have weak end points, a great review cannot come out - As far as possible it is still a good revision - underline the limitations of the endpoints of the studies - restenosis and stent-related outcomes are generally quite weak - emphasize the role of coronary calcium as a clinical marker and underline the growing importance of coronary CT in this regard - Coronary imaging also has its limits in assessing calcium, no matter how "soft" a calcified lesion is. Also for this reason, randomized trials are difficult and the risk of neutral results is high. - Emphasize that there is room for technological improvement in coronary balloons, which are increasingly flexible and small

Author Response

Point 1: Well written review Remarks - Obviously a review collects the strengths and weaknesses of the available studies. If the studies have weak end points, a great review cannot come out - As far as possible it is still a good revision - underline the limitations of the endpoints of the studies - restenosis and stent-related outcomes are generally quite weak. 

Response 1: This is an excellent point and the following sentence has been added to the manuscript: “However it is to be noted that many of these clinical trials lack hard clinical end points and have focused on stent-related outcomes. Additionally, comparision of outcomes across clinical trials should be performed with caution given differences in study population and the inherent heterogeneity in the construct of these calcium modification tools. Finally, further investigation of the results in larger contemporary cohorts is needed.”

Point 2: Emphasize the role of coronary calcium as a clinical marker and underline the growing importance of coronary CT in this regard. 

Response 2: This is a very important point and we added the following sentence to the manuscript: “Coronary CT angiography has emerged as a useful non-invasive tool to identify coronary calcium and plan coronary interventions. Measurement of coronary artery calcium score can be used to stratify cardiovascular risk as it is a powerful predictor of atherosclerotic cardiovascular disease [5].”

Point 3: Coronary imaging also has its limits in assessing calcium, no matter how "soft" a calcified lesion is. Also for this reason, randomized trials are difficult and the risk of neutral results is high. 

Response 3: Thank you for the comment. We have added the following sentence to the manuscript:

“However, it should be recognized that every modality of calcium assessment has its relative advantages and disadvantages.” Additionally we have elaborated on the shortcomings of clinical trials assessing calcium modification techniques (as mentioned under Point 1 above).

Point 4: Emphasize that there is room for technological improvement in coronary balloons, which are increasingly flexible and small.

Response 4: Thank you for this recommendation. The following sentence has been added to the manuscript: “Although modified BA with specialty balloons has advanced in recent years, resulting in better deliverability, there is still opportunity for further improvement in design.”
